# What Makes a Quality Lifestyle Coach? A Theoretical Model Based on the Real-World Context of Delivering the National Diabetes Prevention Program

**DOI:** 10.3390/healthcare13020126

**Published:** 2025-01-10

**Authors:** Lillian H. Madrigal, Olivia C. Manders, Mumta Kadir, Sarah A. Piper, Mary Beth Weber, Linelle M. Blais

**Affiliations:** Rollins School of Public Health, Emory University, Atlanta, GA 30322, USA; olivia.c.manders@emory.edu (O.C.M.); mkadir@umich.edu (M.K.); sarah.ann.piper@emory.edu (S.A.P.); mbweber@emory.edu (M.B.W.); linelle.m.blais@emory.edu (L.M.B.)

**Keywords:** diabetes prevention, health coaching, behavior change, chronic disease prevention, health promotion, health education

## Abstract

Background/Objectives: Lifestyle coaches are integral to delivery of the National Diabetes Prevention Program (DPP); however, few studies have explored the role of the lifestyle coach in relation to participant success. This study aimed to develop a conceptual model of the factors that contribute to lifestyle coach performance and success. Methods: A cross-sectional qualitative study including 82 semi-structured interviews with National DPP staff and participants between June 2020 and February 2022. Results: Based on the analysis of the interviews, the developed model illustrates how central to coach success are the intrinsic qualities they bring to the role, the quality of their training and preparation for the lifestyle coach role, and the mechanization of their qualities and training through their program delivery. Around this focal relationship revolve four other factors that together or independently affect lifestyle coach success: organization influences; external influences; program participants; and evaluation and feedback. Conclusions: Health coaching can be an incredibly powerful tool in behavior change. This model provides insight into how coaching can impact participant outcomes and can be used by other health promotion programs who use the coach model.

## 1. Introduction

Over 130 million adults are estimated to be living with diabetes or prediabetes in the United States (US), resulting in $327 billion in medical costs and lost wages annually [1]. Several randomized controlled trials, including the US Diabetes Prevention Program (DPP), showed that lifestyle interventions focused on weight loss, increasing physical activity, and/or diet change could prevent or delay diabetes in adults with prediabetes [2,3,4,5,6]. Based on the strength of this evidence, the CDC established the National DPP to scale and sustain the implementation of the proven diabetes prevention programs through public and private partner organizations [5,7,8]. The CDC established the Diabetes Prevention Recognition Program (DPRP) which defines national quality standards and practices for delivery organizations and oversees the recognition process to ensure program fidelity across delivery organizations [9]. As of June 2024, there are over 1500 organizations delivering the National DPP in all 50 states, and approximately 780,000 participants have enrolled in the program since 2012 [8]. 

Lifestyle coaches are integral to delivery of the National DPP. Lifestyle Coaches are involved in participant recruitment, retention, and program delivery, and responsible for creating an environment in which participants can make and sustain the positive lifestyle changes needed to achieve their program goals. Lifestyle coach success, measured in terms of their participants achieving National DPP target weight loss goals and overall program retention, is often determined by their implementation approach and interaction with participants [10]. Coaches who are accessible, collaborative, culturally competent, and able to deliver a patient-oriented program have been most influential [10,11]. In general, lifestyle coaching approaches focusing on adherence monitoring and the support of disease-specific, self-management goals, self-efficacy, and commitment, show promise for improving health outcomes [12,13,14]. Additionally, providing a patient-centered curriculum and adjusting the program based on participants’ needs engages members more actively in their treatment, increasing participants willingness and efforts in making lifestyle changes, leading to improvements in outcomes such as weight loss and glycemic improvements [15,16,17]. A 2013 systematic review of health coaching found that there is large variability in the effectiveness of health coaching due to the range of interventions using coaching as well as differences in coaching techniques, theoretical foundations, frequency and duration of coaching process, level of human interaction, the type of content education included, and the professional background and training of coaches [18].

Few studies, however, have explored the perspectives of National DPP program insiders (e.g., lifestyle coaches and program participants) on the role that individual lifestyle coach qualities, practices, and training have on participant success. The QUAL Coach study seeks to fill this gap by presenting the results of qualitative interviews with National DPP lifestyle coaches and participants. These perspectives can provide valuable insights into what characteristics, skills and practices are important in lifestyle coaches, data which can guide both selection and training for the National DPP and similar lifestyle change programs. 

## 2. Materials and Methods

### 2.1. Subjects

Emory Center’s Diabetes Technical Assistance and Training Center (DTTAC) has directly trained over 5000 National DPP lifestyle coaches representing over 2000 organizations across all 50 states and many US territories. A request for interview participants was sent to subscribers of the DTTAC contact list of over 10,000 individuals that deliver the National DPP. Those responding to the call comprised the sampling frame for lifestyle coach, master trainer (coaches who train other coaches), and program coordinator recruitment. Interested participants completed an intake form with self-reported demographics and program delivery information which was used to purposively selected participants to reflect the diversity of implementers by length of time delivering the program, number of cohorts delivered, their organization type and DPRP status, delivery location/region, coach education, background, and other demographic characteristics. To capture the diversity in lifestyle change program participant experience, we asked the lifestyle coach and master trainers we interviewed to nominate current and past program participants that had successfully completed the program, had left the program or were not successful in reaching their goals. We intentionally sought to recruit women under the age of 50, men, and different racial/ethnic backgrounds to be interviewed to increase the diversity of our participant group. Program participants were offered a $25 gift card for their time.

### 2.2. Study Design and Instruments

The QUAL Coach Study Team conducted a cross-sectional qualitative study to gain the perspectives and learn about the experiences of four groups involved with the National DPP: lifestyle coaches, master trainers, organization program coordinators (staff) and the National DPP lifestyle change program participants. Qualitative methods were selected because the study sought to understand perspectives and experiences, focusing on the qualities—the specific characteristics, skills, behaviors, and supportive organizational factors—that these key individuals think contribute to lifestyle coach success. Data were collected via semi-structured interviews between June 2020 and February 2022. This study received approval by Emory University’s Institutional Review Board (#STUDY00000658). 

Semi-structured interview guides were designed from each role’s viewpoint to capture that group’s unique perspectives and experiences relating to each domain (interviewee background; training for the lifestyle coach role; important lifestyle coach competencies; definitions of success; coach instruction practices and approaches; challenges; organizational support and infrastructure; and context and external factors). Questions sought to identify the most salient and impactful aspects of each domain, focusing on identifying important coach characteristics, skills and expertise, practices and approaches, ability to address challenges, and the internal organizational and external factors that impact the work of coaches (Example in Table 1). In this way, we could triangulate responses to tease out any nuances in perspective. A cognitive interview was conducted with a participant from each group that would have been eligible to participate in the study to ensure questions were easy to understand and answer, and that responses aligned with the rationale of the domain and would help the team to answer the research questions. 

### 2.3. Data Collection

The study team conducted 82 interviews. Interviewees were invited to participate in a 60-min interview using Emory’s secure Zoom videoconferencing account; verbal consent and permission to audio-record were obtained prior to initiating the interview. All recordings were transcribed by a third party, quality checked, deidentified, and uploaded into MAXQDA 2020 [19] for coding and analysis. Interviews were conducted until the study team agreed that saturation had been reached for each participant group both in codes and meaning [20]. 

### 2.4. Analysis

The research team developed a deductive codebook—based on conceptual framework domains and interview questions—to code, memo and discuss all transcripts from interviews. To ensure intercoder reliability, all transcripts were coded independently by two members of the research team (L.H.M., O.C.M., M.K., V.P., E.V.); each pair discussed and resolved any differences in coding prior to analysis [21,22]. 

### 2.5. Reflexivity

Data were collected and analyzed by research team members with little to no experience with lifestyle coach training and practices. We believed this would help reduce bias and leading questions during interviewing and in interpretation during analysis. Research team members also questioned each other during code development and coding alignment to reduce possible bias. Additionally, lifestyle coach subject matter experts, QUAL Coach team members at DTTAC, were engaged to provide context and insight to ensure interview questions, coding, and analysis were relevant and meaningful. 

### 2.6. Model Development

The research team met and discussed the code books and key issues identified during coding and analysis to characterize salient factors that interviewees described as predictive or important to lifestyle coach performance and success. The team grouped and categorized these factors and explored their relationships in a series of diagrams. These were discussed and refined together with the programmatic team. Finally, thick descriptions were written for each of the key codes relating to the model that provided the additional insight and details to finalize the model.

## 3. Results

Sixty-one National DPP Staff and 21 National DPP participants were interviewed. Of the staff interviewed, the majority (85%) were lifestyle coaches and many held multiple roles at their organization (Table 2). Around half (49%) of the staff had about 1−3 years of experience implementing the program, 41% had over 4 years of experience and 8% had less than 1 year of experience. Staff delivered the program at different types of organizations. The majority were healthcare settings: healthcare/hospital (34%) and community-based healthcare (20%). Staff were from 38 states and Puerto Rico, and represented rural, suburban, and urban areas fairly equally. The vast majority had some kind of health education or training (public health, nutrition/dietetics, fitness, nursing, pharmacy, etc.). Most staff interviewees identified as women (85%) and white/Caucasian (64%). Similarly, the majority of National DPP participants identified as women (75%) and Caucasian (52%) and around half (48%) were over the age of 65 (Table 3). 

### 3.1. Model Presentation/Overview Description

Based on the interviews, we developed a model of the primary internal and external factors that contribute to lifestyle coach performance and success. Central to coach success are the intrinsic qualities they bring to the role (personal characteristics, skills, habits, experiences), the quality of their training and preparation for the lifestyle coach role, and the mechanization of their qualities and training through their program delivery. Around this focal relationship revolve four other factors that together or independently affect lifestyle coach success: influences of their organization; other external influences; the participants; and evaluation and feedback (Figure 1). 

### 3.2. Lifestyle Coach Performance/Success

National DPP staff were asked how they would know if they were successful in their role as lifestyle coach. The majority of coaches and master trainers first described coach success in terms of more formal metrics, namely by citing the National DPP program goals (type 2 diabetes prevention or delay through achievement of weight loss and physical activity goals). However, success went beyond goal achievement to include less formal metrics, as one coach shared: 


*“outside of that [reaching CDC goals], for me, it’s really hearing the anecdotal information that comes out of the class…People who say, I never thought I could lose weight. I never thought I could exercise like this again. And I think some of that stuff for me is really a success… I think for me it’s really people’s happiness with the outcome that they individually have gotten, I think is the biggest success for me.”*
(Lifestyle Coach 112)

Similarly, many staff participants said that weight-loss should not be the most important outcome that determined participant, coach and program success. They frequently mentioned having participants that did not lose weight but reported benefiting from the program in other ways (increased physical fitness, mental health, social support, etc.). Coaches frequently added that they measure their own success through participant expression of satisfaction and enjoyment of the program, engagement during class, and participant referrals. Some staff participants mentioned feeling successful when they built good rapport with participants as well as when their participants developed good rapport and relationships and provided social support for each other.

### 3.3. Coach Qualities

This domain is defined as the aspects of coach character, personality, personal and professional background, expertise, and skills that contribute to successful outcomes. Across groups, most interviewees discussed aspects of personality or character rather than skills as being integral to participant engagement, buy-in, and retention. They also identified specific skills and qualities of characteristics such as listening/being a good listener as contributing to an environment in which participants could engage and be successful. When asked to describe their master trainers, lifestyle coaches described many of the same qualities and skills valued in coaches detailed below and often tried to emulate or incorporate their master trainer’s practices and qualities in their own program delivery. 

The majority of interviewees across all groups described disposition as the most important quality in a successful coach. Coaches described as “positive” were said to maintain an encouraging, optimistic attitude or outlook throughout the program. They were frequently described as being warm, welcoming, and enthusiastic with participants, and demonstrated empathy, compassion or understanding of the challenges participants face by listening with respect and kindness.

Coaches that shared their own lifestyle changes and health struggles were seen as being open, honest and approachable; qualities that also made them relatable. As one participant said: 


*“You know, I think number one, to me anyway, is somebody I can relate to. Are they personable? And do I really feel like they care about me or are they just doing their job?”*
(Program Participant 409)

Across all groups, the most important skills or expertise that interviewees said a coach should have include the ability to guide behavior change, meet participants where they are, engage participants, create a good group culture/dynamic, build rapport and relationships with and among participants, and establish trust and credibility. When asked about their own or their ideal coaches, most interviewees described them as having a successful presentation or delivery style and as encouraging/supportive to participants. This would help coaches to engage participants, make class content interactive, get participants to speak up, share and participate, and hold the class’s attention. In addition, interviewees shared how important they thought it was for the coach to foster an environment that created a strong group dynamic and relationships through good facilitation and trust-building skills. Many coaches we interviewed said they continually worked to hone their facilitation skills and delivery and presentation style. 

Lastly, many interviewees discussed other aspects of coach encouragement or support of participants in ways that meet their individual needs. This included understanding participant personalities, recognizing their stage in readiness to change, and ability to tailor support in a way that would resonate with each participant. They also described using techniques that support class morale such as making personal notes of encouragement in participant tracking logs, scheduling one-on-one check ins, or providing words of encouragement to the whole group. 

There were differing opinions and no consensus among coaches, trainers, and National DPP participants around what type of professional background is best for a lifestyle coach. Lifestyle coaches and master trainers felt that while it could be helpful for a lifestyle coach to have training in nutrition, physical activity, or health education, many also recognized that it was not mandatory to deliver the DPP effectively. All coaches described both pros and cons of their professional background. Coaches with advanced training said it helped their confidence in delivering program content. However, they also reflected on a need to remember to “put aside my health educator or nurse hat” and refrain from falling back on their training and “telling” or “teaching” participants instead of facilitating and involving them in the learning process. Most coaches felt that the characteristics and skills of a good lifestyle coach were foundational and that being able to engage and relate to participants was more important than formal training or expertise. Similarly, the primary argument some National DPP participant interviewees gave for formal training or credentials was credibility. However, credibility also required that a lifestyle coach have similar life experiences or be relatable; a lifestyle coach with credentials but no personal experience with the DPP or not having had to make lifestyle changes was not credible. For others, credibility had to do with their knowledge of course materials and how well they delivered them. Finally, most National DPP program participants talked about different aspects of a coach’s personality, skills and approach to leading the course as more important than advanced training. 

### 3.4. Quality Training

We defined the quality training domain as National DPP master trainer-led, practice-based lifestyle coach training, using CDC-approved curricula and focusing on facilitation, role modeling, and utilizing tools and resources to create a supportive participant learning environment. Lifestyle coaches described their own experiences receiving training as overwhelmingly positive. Most coaches shared that they liked having experienced-based learning opportunities to develop their facilitation skills, the opportunity to network with other coaches, and hands-on experience with program activities to see how they “work” during training. The practical training structure helped them think about and learn how to break down the curriculum to meet participant needs, facilitate groups and handle tough questions or situations, and create an engaging environment using visuals, activities, and group discussions. Learning activities lifestyle coaches cited as resonating the most included role-playing scenarios; practice teaching sessions; a “grocery bag” activity where food items are pulled from a bag, and trainees discuss their impressions of its nutritional value; and brainstorming the characteristics and attributes of a program participant and lifestyle coach. They emphasized the utility of learning and practicing facilitation skills and techniques in a safe learning environment as key in helping them understand how to promote group participation. 

### 3.5. Quality Delivery

The domain of Quality Delivery was defined as the personal practices, or activities and routines, that lifestyle coaches regularly perform before, during and/or after a class session or cohort to ensure quality and that in some cases contribute to fidelity in program delivery.

To prepare to lead a new cohort, many lifestyle coaches reviewed all program materials to ensure they have the necessary supplies, familiarize themselves with any new content or tools (e.g., food tracking apps), and finalize the schedule and any potential make-up sessions. Many coaches reported the importance of facilitating a discussion on “ground rules” during the initial sessions of a new cohort. They found this helped to establish expectations about how participants will be treated and will treat others as well as fostering trust and a respectful group dynamic. Similarly, most coaches said they habitually review materials before each session as well as notes from the previous session and from the last time they taught the class material. Most coaches mentioned sending email reminders about the session, and arrive early to set up (e.g., materials out, weigh-in station set up, chairs in a circular configuration, music playing). These coaches described the importance of being able to focus on the participants as they arrived, which they said helped create a welcoming and engaging environment for them.

During-class practices were similar with many coaches emphasizing the importance of maintaining a standard routine to provide consistency, which was important to make participants feel comfortable and contributed to establishing expectations. Typically, routines followed a similar format: weigh-in and food tracker collection at the beginning of the class; an ice-breaker activity and check-in with participants to see how they are doing; and then going through the day’s curriculum content. Where coach practices sometimes differed is at the end of the sessions where coaches mentioned working on setting goals, engaging in group physical activity, and/or providing a message of encouragement. 

Coaches used a variety of activities to engage participants with the material, help them understand health information, and encourage cohorts to practice healthy lifestyle behaviors together. Examples include small group discussions, role-playing, demonstrations with visual aids (e.g., measuring cups, props, cooking), incentives (e.g., exercise bands, water bottles, fresh produce/groceries, kitchen tools, wearable fitness technology), engaging in physical activity (e.g., Zumba, hula hoops, exercise videos, walking around the classroom), and use of workbooks. Some coaches and participants mentioned taking field trips to a fitness center or grocery store to engage in hands-on learning. Coaches also talked about being prepared in advance to address a range of participant sociodemographic factors by making sure 


*“our healthy eating examples incorporate examples for those who are shopping at Whole Foods and those who are shopping at Dollar General” and “both those who have a year-long gym membership at […] the elite gym in town and those who are using canned food as their weights at home.”*
(Lifestyle Coach 113)

After a class session, most coaches double check to ensure all participant data was collected, make notes about the curriculum for the next time they deliver the content, schedule make up sessions with absent participants and send any post-class reminders or messages to participants. Coaches use a range of platforms (e.g., Facebook, group chat apps) to encourage participants to stay connected with each other between classes. Additionally, these platforms often help build social support as participants share recipes and lifestyle change tips and organize outside-of-class activities for physical activity or socializing. Coaches encouraged participants to remain connected with their cohort after the program, and coaches or their organization held reunions with graduated cohorts. Coaches also referred participants to other lifestyle change programs sponsored by their organization to help participants maintain health-promoting behaviors. 

Lastly, many coaches made changes in their practice over time as they became more familiar with the curriculum, which helped improve program delivery and outcomes. Coaches reduced their dependency on the curriculum slides, which allowed them to focus more on facilitating conversation and introducing new activities to increase participant knowledge. They also adapted the curriculum materials and examples to fit their participant needs and contexts (e.g., cultural, socio-economic, physical ability, etc.) to assist participants in making appropriate and meaningful behavior changes. Some coaches mentioned the importance of continuously incorporating new engagement strategies to retain participants.

### 3.6. Organizational Influences

Organizational influences includes factors internal to the National DPP delivery organization including leadership, infrastructure, operations, and resources, that may impact the work of the lifestyle coaches positively and negatively. A more in-depth analysis about organizational and external influences has been published elsewhere [23]. Overall, lifestyle coaches and other implementing staff said they have the material resources and organizational leadership support they need to deliver the National DPP. Resources include physical space for classes, materials, virtual delivery platforms, marketing assistance, and incentives. In addition, many implementers felt they had autonomy to make decisions about program implementation. 

Where coaches attributed difficulties to organizational influences, it was largely due to a lack of dedicated staff and/or staff time. Almost all implementing organization staff interviewed had other responsibilities in addition to implementing the National DPP. Many coaches expressed the need for more staff/staff time to implement the program successfully. Some ways in which organizations appeared to alleviate this need and support coaches with delivery was through co-coaching (two coaches assigned to one cohort) and/or having other staff such as nutritionists and fitness instructors regularly attend cohort sessions to provide content expertise. 

### 3.7. External Influences

External influences included factors outside of the delivery organization, including the local and national physical, social, and political environment, external funding mechanisms, partnerships, and referral systems not linked with or incorporated into the organization’s network. When discussing external factors that positively impacted their work, coaches frequently mentioned external partners (e.g., churches, community organizations, academic institutions/student volunteers, healthcare providers, non-profit organizations) that helped recruit participants and host classes. Some coaches felt partnering with organizations serving priority populations (e.g., a Latin American Organization) was very important in successful outreach, recruitment, and assistance in adapting program elements for these groups. Further, many coaches and master trainers mentioned that their local health department had funded their own lifestyle coach and/or master trainer training which illustrates how external entities can directly impact coach training and capacity building.

External factors that hindered program success included lack of key partnerships in the community and challenges with obtaining and maintaining external funding. Many of the organizations rely on outside funding for the National DPP. Several coaches had good funding and technical assistance support from the State or Local Health Department, however there was uncertainty about what would happen to their program if that funding disappeared. 

### 3.8. Program Participants

The program participants domain included demographic factors, challenges, readiness for change and other personal things that participants bring to their participation in the program. Program participants described several factors that could influence participant success with the coach’s ability to identify, address or help a participant work within their particular constraints as key. Race, ethnicity, gender and age, among other lifestyle coach demographic characteristics mattered to some but not all program participants. While some mentioned having someone that looked like themselves instructing the class made them feel more comfortable opening up and relaxing, other participants said the coach’s demographic characteristics mattered less than their personality. One participant shared, 


*“I’m more concerned about someone that can accept people the way they are, […] irrespective of the size, the race, whatever. Just be compassionate with what you do, just be experienced with what you are doing. Just someone that can deliver, someone that has a lot of ideas to share.”*
(Program Participant 420)

Participants often felt that it was more important that the coach was relatable in terms of shared health and wellness histories and life experiences. For example, one participant selected a National DPP lifestyle change program that took place in a church and found that the coach’s spirituality combined with their practices helped them connect with the material. Coaches also reported being very aware of their identity and how that might or might not impact their participants, and took measures to build up trust and to relate to participants. Participant personal motivation or readiness for change was cited by master trainers, coaches, and participants alike as the biggest facilitator or barrier to success for participants regardless of the quality of their coach. 

### 3.9. Evaluation Feedback

The Evaluation Feedback domain included formal and informal forms of feedback given to coaches. While both lifestyle coaches and master trainers desired and collected feedback/evaluations in a variety of ways (e.g., formal evaluations throughout the course, short evaluations at the end), some coaches did not collect formal feedback about either the program or their coaching. Master trainers most frequently mentioned conducting regular formal evaluations at their organizations. In addition to reports submitted to the CDC, some master trainers conducted assessment of knowledge with program participants about behavior change and practices, which was used with participant biometric data to identify lifestyle coaches that might require some additional support. 

Coaches mentioned receiving informal feedback in the form of individualized comments from the participants through emails or one on one conversation. Coaches and master trainers observed participant engagement, participation, and body language, and organically responded by tailoring or modifying their presentation to be more engaging or meet participant needs. Some program participants mentioned observing their coach making these in-class adjustments.

## 4. Discussion

This study revealed rich details about what National DPP staff and program participants think are the important factors influencing coach success in terms of recruitment, retention and participant attainment of biometric goals. This model demonstrates that with the exception of the quality of the training they receive, most of the internal coach factors are either intrinsic qualities of the individual (personality, background experiences) or within their locus of control (preparation, practices). Since National DPP lifestyle coach training prescribes consistency in coaching techniques, fidelity in program delivery, a high-level of participant-focused interaction with both lifestyle coach and peers, the curriculum and coach training (the theoretical foundations of which are grounded in behavioral theory) inherently encourage collaboration and active learning [18,24] and build on lifestyle coach intrinsic qualities. 

External factors are primarily related to recruitment and retention, with participant factors such as readiness for change also influencing participant outcomes. All four interview groups mentioned similar characteristics, skills, and practices they think contribute to successful programmatic and participant outcomes. Further, in many cases, interview participants described a synergy of characteristics, skills and practices that where combined, created an environment in which lifestyle change could take place, which is reflected in this paper’s proposed model. This aligns with the outcomes mentioned in other studies on chronic disease health coaching [13,24] as well as the results of a 2022 systematic review to understand the competencies required for health professionals to deliver patient-centered coaching and improve and optimize care. The competencies identified include effective communication, team and leadership skills, an understanding of evidence-based knowledge, self-awareness of personal gaps in knowledge, lifelong learning, cultural competency, professional behavior and accountability, empathy, confidence, and systematic work, and activities coordination [25]. 

The characteristics described as most important to have in a coach encompassed qualities of character (good listener, positive, warm/welcoming, enthusiastic, empathetic/compassionate, and open/honest) that made participants and lifestyle coach trainees alike feel comfortable and encouraged them to engage with other cohort members and the course materials. The most important skills mentioned included listening, facilitating, guiding behavior change, the ability to “meet participants where they are,” creating a good group culture and dynamic, building rapport and relationships with and among participants, establishing trust and credibility, having a successful presentation or delivery style, and being encouraging and supportive of participants. Interestingly, most of the skills and abilities mentioned above are attributes of facilitation and are dependent on many of the characteristics participants also described as being important. 

Likewise, coach practices such as pre-course preparation and post-course reflection, establishing expectations and routines for consistency, and constant evaluation of participant responses to session materials, all contribute to a confidence in delivering the program, building trust, and creating a comfortable environment. In a similar study on implementing a group-based DPP from the perspective of lifestyle coaches, participants described the ideal lifestyle coach as an engaging group leader that demonstrates excitement and is motivated, prepared, and familiar with the content [10].

Another systematic review including studies of health coaches with predominately medical/healthcare backgrounds identified variability in the effectiveness of health coaching that they attribute to a range of coaching interventions; differences in coaching techniques, theoretical foundations, frequency and duration of coaching process; level of human interaction; type of education content included; and the professional background and training of coaches [18]. While in general our study population follows this trend, our study results and model underscore that the most important factors relating to participant success are independent of a coach’s formal training and background. Factors such as character, skills, personal experience, DPP coaching training and habits may be more important in determining a coach’s success than their formal training and credentials. While training and credentials lend credibility for some National DPP participants, other factors may be more important. Further, the DPP prescribes consistency in coaching techniques and fidelity in program delivery, frequency and duration of coaching over 12 months, a high-level of participant-focused interaction with both lifestyle coach and peers that encourages collaboration and active learning, and theoretical foundations grounded in behavioral theory through its coach training [18,24]. 

Results of this study suggest opportunities for additional research to test different aspects of the model and develop metrics to quantify recommendations. This includes developing lifestyle coach instruction focusing on hands-on practice to develop and reinforce the skills and practices that stakeholder groups identified as being essential in a lifestyle coach and developing guidance for recruiting and identifying DPP lifestyle coach candidates possessing the intrinsic qualities linked to successful participant and programmatic outcomes. This would include measuring coach qualities, training, and practices in relation to coach outcomes, and would require testing and refinement of both guidance and instruction. This model could be used to develop and test metrics for coach recruitment based on the internal factors identified and defined, as well as additional coach preparation to address external factors that influence coach success, including sources of support for the program, lifestyle coach and program needs, and troubleshooting and addressing implementation barriers and facilitators. The model could also be used as a tool for identifying root causes of challenges in initial coach training relating to external factors such as organizational support, opportunities for program delivery, and/or as a model for coach evaluation.

### Strengths & Limitations

Strengths of the study include diversity in representation of the sample which allowed us to compare experiences and perspectives by and across user groups. Participants were purposively selected from across program types and roles as well as geographic location to ensure representation in experience. Steps were taken throughout the study to ensure academic rigor, including pre-test of interview guides with content experts to ensure relevance and clarity of questions; standardization in interviewing methods through extensive interviewer training and prescribed procedures, and development of deductive and inductive codes to capture standardized factors in implementation research (deductive) as well as themes and concepts that emerged in the data (inductive). 

Limitations include biases that may have been introduced through the method used to generate the sampling frame which was comprised of individuals registered on the DTTAC listserv that volunteered to participate. Similarly, we were unable to recruit participants who dropped out of the National DPP or did not reach study goals. This may have established a positive bias towards the program due to participation in cohorts led by successful coaches that provided high quality experiences and whose participants had good results. Additionally, there is limited representation of the experiences of lifestyle coaches and master trainers that received training outside of DTTAC.

## 5. Conclusions

Health coaching can be an incredibly powerful tool in health behavior change. Our model of the National DPP lifestyle coach provides insight into how the intrinsic characteristics and qualities of individuals selected to become coaches together with how they are trained can impact positively participant outcomes and can be used by other health promotion programs who use the coach model. 

## Figures and Tables

**Figure 1 healthcare-13-00126-f001:**
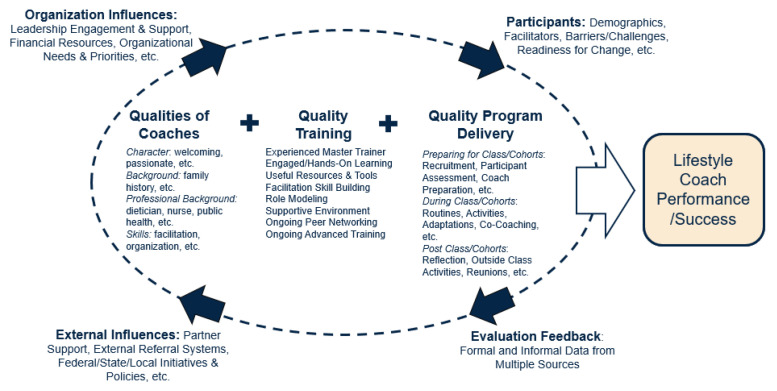
QUAL Coach Theoretical Model.

**Table 1 healthcare-13-00126-t001:** Interview Guide Example Questions & Probes.

Domain: Types of Information Sought	Lifestyle Coach Questions	Master Trainer Select Questions	National DPP Participant Questions
**Competencies:** To learn what competencies (skills/expertise) coaches find most valuable, their self-efficacy with these competencies, and if the initial training they received provided them with sufficient competency in these areas.	What skills or expertise gained do you think were most important to develop during your training? Generally, what would you consider the top three lifestyle coach skills for facilitating the life change program groups? How confident are you in your ability to use these skills with the groups you lead? In what ways, if any, have you continued to build LC skills or expertise since your training? Probe: Have you received any additional development of these skills since the initial training? Formally or informally- on the job	What important skills or expertise did you have an opportunity to develop during the lifestyle coach training? What additional skills or expertise did you gain from your MTS training that you have found most helpful/applicable to your work as a lifestyle coach? Generally, what would you consider the top three MT skills for training lifestyle coaches? How confident are you in your ability to use these skills when you train lifestyle coaches? In what ways, if any, have you continued to build your master trainer skills or expertise since your training? Probe: Have you received any additional development of these skills since the initial training? Formally or informally- on the job	As you reflect back on your experience with the program what stood out to you most about your lifestyle coach? Probe: This might include things like personality, attitude, facilitation style etc. What do you think are the most important skills or areas of expertise for a lifestyle coach to have? How would you describe that group experience? Probe: Did your coach do anything specifically to nurture/create that group environment? What could the coach have done to strengthen the group experience?

Note: The competencies domain was not included in the organization program coordinator guide, which was focused on internal and external factors related to implementation at the organization level.

**Table 2 healthcare-13-00126-t002:** National DPP Staff Interviewees (N = 61).

Role ^a^	N	%	Education Background/Training	N	%
Lifestyle Coaches	52	85%	Public Health	16	26%
Master Trainers	21	34%	Nutrition/Dietetics	14	23%
Program Coordinators	29	48%	Fitness/Physical Activity	8	13%
			Nursing	5	8%
**Years implementing the program**			Other Health Field	4	7%
Less than 1 year	5	8%	Pharmacy	3	5%
1−3 years	30	49%	Other	8	13%
4−6 years	16	26%	missing	3	5%
7+ years	9	15%			
missing	1	2%	**Gender**		
			Man	7	11%
**Organization DPRP Status**			Woman	52	85%
Fully Recognized	28	46%	missing	2	3%
Pending/Preliminary	30	49%			
Lapsed/No longer active in the DPRP	1	2%	**Race/Ethnicity**		
missing	2	3%	White/Caucasian	39	64%
			Black or African American	9	15%
**Organization Type**			Asian	5	8%
Healthcare/Hospitals	21	34%	More than one race	3	5%
Community-based healthcare	12	20%	American Indian or Alaskan Native	2	3%
Community-based organizations	7	11%	Other: Puerto Rican	1	2%
Government agencies	8	13%	missing	2	3%
Other: Health insurers, Employers, Academia	13	21%			
			**Hispanic/Latino**		
**Location/Urbanicity**			No	52	85%
Rural	21	34%	Yes	7	11%
Suburban	19	31%	missing	2	3%
Urban	19	31%			
missing	2	3%	**Age**		
			25−34 years	13	21%
			35−44 years	13	21%
			45−54 years	14	23%
			55−64 years	16	26%
			65 years or older	1	2%
			missing	4	7%

^a^ Interviewees could have more than one role

**Table 3 healthcare-13-00126-t003:** National DPP Participant Interviewees (N = 21).

Gender	N	%	Age		
Man	5	24%	35−44 years	1	5%
Woman	16	76%	45−54 years	5	24%
			55−64 years	5	24%
			65 years or older	10	48%
**Race/Ethnicity**			**Hispanic/Latino**		
White/Caucasian	11	52%	No	20	95%
Black or African American	9	43%	Yes	1	5%
American Indian or Alaskan Native	1	5%			

## Data Availability

De-identified qualitative data is available by reasonable request to the corresponding author. Quantitative data was from DTTAC program records which are not publicly available.

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
