# Peer review of "What Makes a Quality Lifestyle Coach? A Theoretical Model Based on the Real-World Context of Delivering the National Diabetes Prevention Program"

_healthcare, 2025, doi:10.3390/healthcare13020126_

Round 1
Reviewer 1 Report
Comments and Suggestions for Authors
This qualitative study evaluated the qualities, training and expertise, behaviors, and factors of National Diabetes Prevention Program (DPP) lifestyle coaches which facilitate participant success. Data was received from a variety of program insiders and participants to address a lack of first-hand perspectives on coaching characteristics for success in the National DPP. This manuscript is well-written, including an introduction which effectively sets up for the rest of the manuscript, detailed methods and results, and a well-thought-out discussion. Some very minor corrections would be needed for this manuscript to be ready for publication.
Abstract: DPP abbreviation needs to be established in the first sentence
Introduction: Page 2 line 53: remove the unnecessary comma after 2013
Methods: The results present demographic information on DPP staff and previous participants, but it is not described in the methods how this information is collected. Most important to note is that it appears to be self-reported data
Results: Page 7 line 197: there appears to be an out of place parentheses
Tables: Table 2 – there is information on location/urbanicity, but I don’t see anywhere in the manuscript how this was classified. Was this self-reported based on the participants perception of their urbanicity? And was it where they live or where they deliver the DPP?
Author Response
Comment 1: This qualitative study evaluated the qualities, training and expertise, behaviors, and factors of National Diabetes Prevention Program (DPP) lifestyle coaches which facilitate participant success. Data was received from a variety of program insiders and participants to address a lack of first-hand perspectives on coaching characteristics for success in the National DPP. This manuscript is well-written, including an introduction which effectively sets up for the rest of the manuscript, detailed methods and results, and a well-thought-out discussion. Some very minor corrections would be needed for this manuscript to be ready for publication.
Abstract: DPP abbreviation needs to be established in the first sentence
Response 1: Thank you for your thoughtful review and comments on our paper. An abbreviation has been added on line 10.
Comment 2: Introduction: Page 2 line 53: remove the unnecessary comma after 2013
Response 2: Comma was removed on line 53.
Comment 3: Methods: The results present demographic information on DPP staff and previous participants, but it is not described in the methods how this information is collected. Most important to note is that it appears to be self-reported data
Response 3: Thank you for pointing this out. On line 74 we have included additional information, “Interested participants completed an intake form with self-reported demographics and program delivery information which was used to purposively selected participants to reflect the diversity of implementers by length of time delivering the program, number of cohorts delivered, their organization type and DPRP status, delivery location/region, coach education, background, and other demographic characteristics (Table 2).”
Comment 4: Results: Page 7 line 197: there appears to be an out of place parentheses
Response 4: Parentheses was deleted.
Comment 5: Tables: Table 2 – there is information on location/urbanicity, but I don’t see anywhere in the manuscript how this was classified. Was this self-reported based on the participants perception of their urbanicity? And was it where they live or where they deliver the DPP?
Response 5: Yes, this was self-reported. In the above sentence starting on line 74, we have added “delivery” to location/region. We asked program staff to report what kind of location “urban, suburban, rural” their programs are located in.
Reviewer 2 Report
Comments and Suggestions for Authors
Congratulations to the authors on completing this manuscript. This study provides an analysis of the success factors of lifestyle coaches within the National Diabetes Prevention Program (National DPP) and proposes a theoretical model that explores the internal and external influences on coach performance. The study has significant practical value but also requires further revisions and refinements. Below are specific suggestions:
Unaddressed Link Between Health Interventions and Inflammatory Mechanisms: While the manuscript discusses how lifestyle coaches help achieve weight management and behavior change, it does not address the potential biological mechanisms, such as inflammation and metabolic pathways. It is suggested that the authors discuss the associations between lifestyle factors (insufficient sleep, sedentary behavior, and unhealthy diet) and health outcomes, particularly the potential influence on inflammatory markers. For example, previous studies have identified associations between sleep, dietary intake, and inflammatory biomarkers, as well as the regulatory role of physical activity. Referencing these studies would help to enhance the theoretical scope and biological explanations in the manuscript.
In the Introduction, the authors could expand the background discussion on the connection between health behaviors, inflammation, and biological mechanisms, thus broadening the theoretical foundation of the study. For instance:
“Recent studies have shown that insufficient sleep is associated with health risks through inflammatory biomarkers and cellular pathways, while moderate participation in physical activity plays a critical regulatory role in this relationship [doi: 10.1111/cns.14783]. These findings provide a biological basis and practical significance for health behavior interventions, such as those promoted by lifestyle coaches through physical activity and healthy eating.”
In the Discussion section, specifically under 3.3 Coach Qualities where coaching skills are discussed, the authors can add content to emphasize the role of physical activity in health promotion:
“By guiding and encouraging moderate physical activity, coaches can help participants not only achieve weight management but also reduce inflammation, thereby improving health outcomes. Recent real-world studies found that recreational physical activity significantly mediates the relationship between dietary intake, systemic immune-inflammatory indices, and depressive symptoms, further supporting the effectiveness of such intervention strategies [doi: 10.3390/nu16060777; doi: 10.1139/apnm-2023-0550].”
The Discussion section can further highlight the effects of lifestyle interventions (e.g., physical activity and behavior changes) on reducing inflammatory biomarkers and mitigating health risks:
“Studies have shown that accelerometer-measured physical activity and sedentary behavior are closely associated with C-reactive protein levels in individuals with insufficient sleep. Replacing sedentary time with physical activity has been shown to effectively reduce inflammation levels [doi: 10.1080/02640414.2024.2348906]. This suggests that lifestyle coaches not only facilitate behavior changes to help participants achieve program goals but also play an important role in addressing health risks associated with insufficient sleep and inflammation, thus enhancing the long-term impact of health interventions.”
Lack of Validation and Quantification of the Model: While the current study proposes a theoretical model based on qualitative interviews, it lacks quantitative data or empirical validation to support its effectiveness. The authors are encouraged to acknowledge this limitation and propose directions for future quantitative validation.
Minor Suggestions:
1.Structure and Logic: The authors could consider combining the “Coach Qualities” and “Training Quality” sections to improve the internal logic and flow of the manuscript.
2.References: Some cited references are outdated (e.g., reviews from 2013 and 2014). It is recommended to incorporate more recent studies to ensure the timeliness of the theoretical background. Moreover, sleep, as an essential component of a healthy lifestyle, is underrepresented in this manuscript. Supplementing the discussion with mechanistic insights will significantly enhance the study’s depth and research significance.
3.Figure Visualization: While Figure 1 (the model diagram) is clear, it could be further simplified or enriched with practical examples to help readers better understand its real-world application.
Author Response
Comment 1: Congratulations to the authors on completing this manuscript. This study provides an analysis of the success factors of lifestyle coaches within the National Diabetes Prevention Program (National DPP) and proposes a theoretical model that explores the internal and external influences on coach performance. The study has significant practical value but also requires further revisions and refinements. Below are specific suggestions:
Unaddressed Link Between Health Interventions and Inflammatory Mechanisms: While the manuscript discusses how lifestyle coaches help achieve weight management and behavior change, it does not address the potential biological mechanisms, such as inflammation and metabolic pathways. It is suggested that the authors discuss the associations between lifestyle factors (insufficient sleep, sedentary behavior, and unhealthy diet) and health outcomes, particularly the potential influence on inflammatory markers. For example, previous studies have identified associations between sleep, dietary intake, and inflammatory biomarkers, as well as the regulatory role of physical activity. Referencing these studies would help to enhance the theoretical scope and biological explanations in the manuscript.
In the Introduction, the authors could expand the background discussion on the connection between health behaviors, inflammation, and biological mechanisms, thus broadening the theoretical foundation of the study. For instance:
“Recent studies have shown that insufficient sleep is associated with health risks through inflammatory biomarkers and cellular pathways, while moderate participation in physical activity plays a critical regulatory role in this relationship [doi: 10.1111/cns.14783]. These findings provide a biological basis and practical significance for health behavior interventions, such as those promoted by lifestyle coaches through physical activity and healthy eating.”
In the Discussion section, specifically under 3.3 Coach Qualities where coaching skills are discussed, the authors can add content to emphasize the role of physical activity in health promotion:
“By guiding and encouraging moderate physical activity, coaches can help participants not only achieve weight management but also reduce inflammation, thereby improving health outcomes. Recent real-world studies found that recreational physical activity significantly mediates the relationship between dietary intake, systemic immune-inflammatory indices, and depressive symptoms, further supporting the effectiveness of such intervention strategies [doi: 10.3390/nu16060777; doi: 10.1139/apnm-2023-0550].”
The Discussion section can further highlight the effects of lifestyle interventions (e.g., physical activity and behavior changes) on reducing inflammatory biomarkers and mitigating health risks:
“Studies have shown that accelerometer-measured physical activity and sedentary behavior are closely associated with C-reactive protein levels in individuals with insufficient sleep. Replacing sedentary time with physical activity has been shown to effectively reduce inflammation levels [doi: 10.1080/02640414.2024.2348906]. This suggests that lifestyle coaches not only facilitate behavior changes to help participants achieve program goals but also play an important role in addressing health risks associated with insufficient sleep and inflammation, thus enhancing the long-term impact of health interventions.”
Response 1: Thank you for your comments, while we appreciate the science connecting lifestyle changes to inflammatory mechanisms, the National DPP has not studied this connection as key outcome and therefore we do not believe this is an appropriate link to emphasize for this paper. Additionally, our focus is on the lifestyle coach factors, not the outcomes of program participants.
Comment 2: Lack of Validation and Quantification of the Model: While the current study proposes a theoretical model based on qualitative interviews, it lacks quantitative data or empirical validation to support its effectiveness. The authors are encouraged to acknowledge this limitation and propose directions for future quantitative validation.
Response 2: Yes, we certainly agree with this limitation. In line 479-494 we discuss the need to additional quantitative research based on this initial model.
Minor Suggestions:
Comment 3: 1.Structure and Logic: The authors could consider combining the “Coach Qualities” and “Training Quality” sections to improve the internal logic and flow of the manuscript.
Response 3: We have reviewed these sections and will continue to use the two sections to ensure clarity between these domains in our model.
Comment 4: 2.References: Some cited references are outdated (e.g., reviews from 2013 and 2014). It is recommended to incorporate more recent studies to ensure the timeliness of the theoretical background. Moreover, sleep, as an essential component of a healthy lifestyle, is underrepresented in this manuscript. Supplementing the discussion with mechanistic insights will significantly enhance the study’s depth and research significance.
Response 4: Yes, we agree that sleep is an essential component of a health lifestyle, however the National DPP does not currently focus on this outcome in their lifestyle change program and therefore it is not a focus of this paper.
Comment 5: Figure Visualization: While Figure 1 (the model diagram) is clear, it could be further simplified or enriched with practical examples to help readers better understand its real-world application.
Response 5: In order to keep the diagram simple we have included practical examples in the narrative throughout the results section.
Round 2
Reviewer 2 Report
Comments and Suggestions for Authors
Good revision